

# Morphology and Mixing of BC Particles Collected in Central California During the CARES Field Study

**Ryan C. Moffet[1], Rachel E. O'Brien[1,2,#], Peter Alpert[3,*], Stephen T. Kelly[2,^], Don Q. Pham[1], Mary K. Gilles[2], Daniel Knopf[3], Alexander Laskin[4]**

[1]Department of Chemistry, University of the Pacific, Stockton, 95211, United States of America
[2]Chemical Sciences Division, Lawrence Berkeley National Laboratory, Berkeley, 94720, United States of America
[3]Institute for Terrestrial and Planetary Atmospheres, School of Marine and Atmospheric Sciences, State University of New York, Stony Brook, United States of America
[4]Pacific Northwest National Laboratory, W. R. Wiley Environmental Molecular Sciences Laboratory

#Present address: Department of Civil and Environmental Engineering, Massachusetts Institute of Technology, Cambridge, Massachusetts 02139, USA
*Present address: CNRS, UMR5256, IRCELYON, Institut de Recherches sur la Catalyse et l'Environnement de Lyon, Villeurbanne F-69626, France
^Present address: Carl Zeiss X-ray Microscopy Inc., Pleasanton, CA, 94588 USA

*Correspondence to*: Ryan C. Moffet (rmoffet@pacific.edu)

**Abstract.** Aerosol absorption is strongly dependent on the internal heterogeneity (mixing state) and morphology of individual particles containing black carbon (BC) and other non-absorbing species. Here, we examine an extensive microscopic data set collected in the California central valley during the CARES 2010 field campaign. During a period of high photochemical activity and pollution buildup, the particle mixing state and morphology were characterized using Scanning Transmission X-ray Microscopy (STXM) at the carbon K-edge. Observations of compacted BC core morphologies and thick organic coatings at both urban and rural sites provide evidence of the aged nature of particles. Based on the observation of thick coatings and more convex BC inclusion morphology, the contribution of fresh BC emissions at the urban site was relatively small. These measurements of BC morphology and mixing state provide important constraints for the morphological effects on BC optical properties expected in aged urban plumes.

## 1 Introduction

Aerosols have a direct effect on climate by scattering and absorbing solar radiation. Black carbon (BC), which results from the incomplete combustion of hydrocarbons from a variety of fuels, absorbs solar radiation across the visible spectrum resulting in a warming effect (Bond et al., 2013;IPCC, 2013). BC is estimated to be the second most potent atmospheric warming agent with carbon dioxide being the most potent. Reducing BC emissions would rapidly counteract the heating by greenhouse gases (Ramanathan and Xu, 2010;Ramanathan and Feng, 2008;Rogelj et al., 2014). To predict the amount of cooling by removing BC from the atmosphere, the direct effect due to BC must be modeled using realistic assumptions.

Because of the insufficient knowledge of particle morphology, mixing state, and interactions with other atmospheric constituents, the radiative effect of BC is uncertain. Radiative transfer models require knowledge about scattering and



absorption cross sections and the angular dependence (phase function) of scattered light. Estimates of these parameters are obtained using Mie theory which assumes particles are homogenous spheres. However, aerosols containing BC are frequently internally mixed. Hence, within the same particle, the BC is heterogeneously mixed with non-absorbing species (Adachi and Buseck, 2008;Moffet et al., 2013). To address the heterogeneous particle structure, the BC has been modeled as

a centrally located sphere evenly coated by non-absorbing material (Ackerman and Toon, 1981). Implementing this type of core shell-theory into global modeling studies suggests a large warming due to the internal mixing of BC (Jacobson, 2001). Numerous recent studies indicate that assuming that the BC is located exactly in the center of the particle over estimates BC absorption (Adachi et al., 2007;Cappa et al., 2012). Several studies point out that a lower BC absorption is obtained by offsetting the BC inclusion from the center towards particle edge, recent field studies confirm that the core-shell model may

not be valid and that other modeling approaches are needed (Cappa et al., 2012;China et al., 2013). Alternate approaches include the Maxwell-Garnett approximation as well as the discrete dipole approximation (DDA) (Scarnato et al., 2013). The Maxwell-Garnett is an "effective medium" approximation whereby an effective dielectric constant is calculated using the dielectric constants of the host and the inclusion. Effective medium approaches still use Mie theory to generate cross sections and phase functions. Alternatively, DDA uses an array of dipoles with prescribed optical properties to calculate cross

sections and phase functions. If enough dipoles are used with DDA, the calculation theoretically becomes exact but is computationally expensive. To test the validity of these theoretical approaches on calculating aerosol radiative properties, detailed morphological and chemical measurements are required at the individual particle level.

In this study, Scanning Transmission X-Ray Microscopy (STXM) is used to quantify the morphology and mixing state of BC-containing particles collected from the Carbonaceous Aerosols and Radiative Effects Study (CARES). During the June

2010 CARES study, a comprehensive set of aerosol, gas, and meteorological parameters were measured (Moffet et al., 2013;Zaveri et al., 2012;Fast et al., 2012). The CARES field study focused on the chemical and physical properties of organic carbon and BC containing particles, and several reports have examined the characteristics of BC particles (Cappa et al., 2012;Cahill et al., 2012;Cazorla et al., 2013;Chakrabarty et al., 2014). Thus far, none report direct measurements of the chemical and morphological properties for a statistically significant number of BC particles. This report presents a

microscopic analysis for a large number of BC particles (~1900 BC containing particles out of a data set containing ~20,000 particles) collected during selected dates of the CARES field campaign. This manuscript utilizes and builds upon the dataset presented in two earlier studies (Moffet et al., 2013;O'Brien et al., 2015). Results presented here can be used to validate assumptions employed in optical and radiative transfer models.

## 2 Experimental

Samples of atmospheric particles were collected during the CARES campaign conducted in June 2010 in the Central Valley, California. Field sites and sampling procedures are described in previous publications (Moffet et al., 2013;Zaveri et al., 2012) and are only briefly described here. Sampling occurred at two primary field sites: the first site was in the Sacramento





urban area (T0 site) expected to have enhanced fresh emissions, and the second site was located 40 km east of T0 in the Sierra Nevada foothills (T1 site) expected to have enhanced aged aerosol. Samples utilized here were collected over two days (June 27 and June 28) during a period of high temperatures and increased aerosol loadings over T0. Airmasses from T0 were expected to be transported over the T1 site (Moffet et al., 2013;Fast et al., 2014). Aerosols were collected onto several

substrates including Si wafers for ice nucleation studies (Knopf et al., 2014), Si wafers containing $Si_3N_4$ membrane windows (Moffet et al., 2013) and Formvar/Carbon type B coated copper grids (Ted Pella, Redding, CA) using a time resolved aerosol collector (TRAC) (Laskin et al., 2006;Laskin et al., 2003). After collection, the samples were sealed and stored at ambient temperature (~21°C) and relative humidity (~50%). Sealing the samples prevented additional exposure to light and relative humidity.

STXM analysis was performed continuously between from 2010 to 2015 at Lawrence Berkeley National Laboratory's (LBNL) Advanced Light Source (ALS) at beamline 5.3.2.2 described in detail elsewhere (Kilcoyne et al., 2003). The STXM instrument provides raw images with photon counts representing pixel intensities. The pixel intensities are converted to optical density (OD) by the relation $\ln \frac{I}{I_0} = -\mu \rho t$ where $I_0$ and I are the photon counts for particle-free and particle-containing regions of the image respectively, μ is the mass absorption coefficient and t is the particle thickness. MATLAB

algorithms described originally in an earlier publication (Moffet et al., 2010) were used to identify the regions of an aerosol containing BC, organic carbon, and inorganic species. However, the mapping algorithms implemented here utilized only four images at different photon energies. Mapping with four images decreases analysis time, allowing for higher throughput and thus, analysis of a larger population of particles. To generate a carbon-based map, aerosol particles were imaged at 278 eV (the carbon pre-edge), 285.4eV ($sp^2$ C*=C), 288.6eV (C*OOH), and 320 eV (the carbon post-edge). Characteristic single

energy images at these energies are shown in **Figure 1A-D**. Typically, at each energy, a 15 x15 μm$^2$ image was acquired with 0.035 μm pixel size and 1 ms dwell time. Occasionally, ~120 different constant energy images were utilized in this study to obtain a high resolution carbon spectrum. For consistency, the set of four constant energy images were used to classify particles for both the 120 energy image spectra and 4 energy image spectra.

Maps derived from these four images are shown in **Figure 1 E-G**. To map "organic" regions, the difference between the

carboxylic (C*OOH, 288.6 eV) and the pre-edge of carbon is used and the resulting map is shown in **Figure 1E**. The inorganic map, representing non-carbonaceous inorganic dominant regions, is derived from the ratio of the pre edge (278 eV) to the post-edge (320 eV) as described earlier (Moffet et al., 2010) and is shown in **Figure 1F**. BC is mapped by normalizing the C*=C $sp^2$ hybridized carbon peak at 285.4 eV to to the post-edge absorbance at 320 eV. This ratio is then scaled with respect to highly oriented pyrolitic graphite (HOPG) enabling calculation of the $sp^2$ hybridization fraction

(Hopkins et al., 2007):

$$\%sp^2 = \frac{OD(285.4 \, eV)}{OD(320)} \times \frac{OD(320)_{HOPG}}{OD(285.4)_{HOPG}}$$





BC particles, which mostly consist of elemental carbon, are expected to have a high %sp$^2$ (>35%) and appear as bright areas in Figure 1G. This paper focuses on quantifying the morphology of particles containing these high %sp$^2$ regions which are defined as BC.

Each of the three maps in **Figure 1E-G** were set using the following criteria: 1) pixels at 288.6 eV with intensities three times below signal to noise ratio were excluded 2) the pre-edge to post-edge ratio was set above a value of 0.5 as discussed in Moffet et al. 2010 and 3) the %sp$^2$ was set above a value of 35% as discussed elsewhere (Moffet et al., 2011). Areas of each of the maps with fewer than 7 conjoined pixels were excluded. In **Figure 1H**, to highlight the BC mixing state these three maps are overlaid in the following order: organic, inorganic, sp$^2$.

Morphological information for the compositional regions of interest (inclusion center of mass, convexity, area) was obtained using the MATLAB image processing toolbox and other custom-written algorithms. Particles on the edge of a field of view were not included in the analysis. Individual particle maps were cropped and stored for interactive and query based plotting. Interactive single particle maps were utilized for quality control of the data and to exclude particles that were misidentified as BC. For the interactive visualization, all of the particles within a user defined subset are displayed. Individual particles are selected via a graphical user interface. Upon selection, the raw stack data for that particular dataset are activated to allow further scrutiny of the data. Occasionally, visual inspection indicated particles may have been misclassified. These particles were omitted from the analysis. Specific biological particle types were identified erroneously as BC and were excluded from the dataset based on their morphology and/or spectral characteristics. Additionally, small nominally pure BC particles were occasionally unidentified due to the specifics of the initial particle detection methods. Another problem area for the identification of BC particles are regions close to inorganic inclusions due to high background levels caused by non-carbonaceous species. MATLAB routines packaged as an application are available at https://www.mathworks.com/matlabcentral/fileexchange/58259-stxm-particleanalysis2-gui.

## 3 Results

### 3.1 BC Mixing State

X-ray spectromicroscopy is one of the few techniques that use molecular markers for imaging the internal structures of BC containing particles. **Figure 1** shows a typical field of view for samples collected during the CARES campaign. The BC map in **Figure 1G and 1H** demonstrates a variety of BC morphology including large, fractal particles, large compact particles and small compact particles. BC particles were internally mixed with inorganic and organic material. For example, the large fractal BC particle (particle 1, **Figure 1G and 1H**) has a small compact inorganic inclusion on its upper extremity and is surrounded by organic carbon. Other BC containing particles have small compact BC inclusions located towards the center of the particle (particles 3 and 4). Some of the small BC inclusions are surrounded by mostly organic material (particle 4) whereas other small compact BC particles are surrounded by inorganic materials (particles 5 and 6). The organic map shown in **Figure 1E** demonstrates that there are organic coatings found surrounding most of the particles. Furthermore, the



branched BC particle shows considerable intensity in the organic (C*OOH). The magnitude of the ratio of the pre edge to the post edge is proportional to the total mass of non-carbonaceous inorganic species. Inorganic inclusions appear as regions of high intensity in **Figure 1F** or blue areas in **Figure 1H**. Sea salt particles are common in the region and based on their cubic morphology, the larger particles containing inorganic regions are likely sea salt. Smaller particles having inorganic dominant regions likely contain sulfate (Moffet et al., 2013).

Based on these maps, the mixing state (OCBC, OCBCIN, OC, IN, INOC) of the particles was stored in a database for the subsequent analysis of BC morphology. **Figure 2** shows cropped single particle maps for all BC particles utilized in this study separated by the sampling site and size (sub and super micron). The most striking difference between BC particles from T0 and T1 is the high amount of inorganic dominant regions for T0 particles. T0 was impacted by sea spray and sulfates from petroleum refineries located in the San Francisco Bay Area. The large inorganic dominant particles can be attributed to sea spray particles that have coagulated with BC emissions from the Bay Area. Many of the smaller inorganic dominant particles are likely agglomerates of BC and sulfates. Indeed, many of the inorganic regions for submicron particles have elongated inorganic regions which are consistent with the crystalline structure of ammonium sulfate (Li et al., 2003). Frequently, BC inclusions were seen on the outside edge of an inorganic dominant region; this arrangement may have occurred upon efflorescence, when a salt excludes the aqueous phase (Liu et al., 2008). At T1, the majority of the BC containing particles included a larger fraction of organic dominant regions as a result of the increased photochemical age and large availability of secondary organic aerosol precursors in the foothills of the Sierra Nevada mountains (Moffet et al., 2013) .

A small portion of particles contained more than a single BC inclusion. In some cases, the identification of more than one BC inclusion was determined to be an artifact of the automated analysis used in the identification of the BC inclusions. However, in many cases, larger particles tend to be associated with more than one BC inclusion per particle as confirmed manually with an interactive version of **Figure 2**. Few studies have examined the radiative effects of particles containing multiple BC inclusions. Others (Jacobson, 2006) have commented that as hydrometeors (cloud and precipitation particles) evaporate, the non-volatile BC inclusions coalesce. While submicron aerosol may not be necessarily identical to hydrometers discussed in that study, it is expected that the behavior upon evaporation is similar. Here, some particle images show that the BC inclusions are separated by inorganic (presumably crystalline) dominant regions.

For the entire particle population, particle mixing states and the fraction of BC particles were characterized for each sampling time as indicated in **Figure 3**. **Figure 3** shows the number of particles analyzed and the fraction of those particles with a particular mixing state. Based on the individual particle maps OCBC and OCBCIN, particles were distinguished from OC and INOC. Overall, the major difference between T0 and T1 is the larger abundance of nominally pure homogenous organic particles at T1. As seen in **Figure 2**, BC particles at T1 are more frequently mixed with the OC phase rather than with the IN phase. Generally, as the number of nominally pure OC particles increased, the OCBC particle type increased ($r = +0.50$, $r^2=0.25$). Additionally, as the IN particle type increased the OCBCIN particle type increased ($r = +0.79$, $r^2=0.62$). As can be qualitatively seen in **Figure 2**, most of the carbon mass likely comes from the particle BC component. This





observation is supported by the mass based carbonaceous mixing state of a smaller subset of this data parameterized using mass based entropy metrics (O'Brien et al., 2015). Generally, due to the dense nature of the BC carbon, emission of BC particles controls trends in the overall carbonaceous mixing state. Given that the average individual particle diversity did not increase with the bulk population diversity, the BC containing particles are considered to be externally mixed with respect to

the carbonaceous mixing state defined using STXM measurements (O'Brien et al., 2015). Nevertheless, the majority of BC containing particles are associated with other species. Because the size and morphological characteristics of the OC and IN phases within the BC particles are expected to impact the optical properties of the particles, those properties are quantified here.

## 3.2 Size and Shape Characteristics

For internally mixed BC particles, both the size of the BC inclusion and the overall size of particle have the greatest influence on the light extinction (Moffet and Prather, 2009). Most importantly, the absorption cross section is driven largely by the size of the BC inclusion. To provide a best estimate of the BC inclusion size distribution and overall particle size distributions, the identified BC areas were used to calculate a circular equivalent diameter ($D_{eqiv} = 2\sqrt{\frac{A_{roi}}{\pi}}$), where $A_{roi}$ is the area of the region of interest (ROI) identified by the mapping described above. **Figure 4a** indicates that the total host particle

size characteristics of BC containing particles sampled at T0 and T1 scaled by the transmission efficiency of the impactor (see supplementary information). These characteristics are similar, but there are small differences that may be due to particle aging. Compared to T0, the total particle distribution for T1 shows a significantly higher population of larger particles. This result is expected considering that the T0 site is located in a source region for freshly emitted BC particles. Due to growth by condensation of organic material, larger particle sizes are also expected at T1. Distributions from both sites show enhanced

numbers in the droplet mode (above 250 nm). The transmission efficiency curve shown in **Figure S1** systematically overestimates the transmission for particles below 300 nm. Size distributions obtained using a scanning mobility particle sizer show much higher populations below ~100 nm, falling off more sharply than the total particle size distributions obtained here (Zaveri et al., 2012). Nevertheless, the influence of coating on the particles is apparent; we attribute the coating to the condensation of organic material on BC particles at the T1 site where particles are expected to be more aged.

The single scattering albedo of BC particles is highly sensitive to the size of the BC inclusion (Moffet and Prather, 2009). The fact that BC inclusion sizes at T0 and T1 are similar with only minor differences (**Figure 4b**), suggests that restructuring of the particles to more compact shapes upon transport is negligible. Previous studies have implied that as BC particles age, the morphology changes from a branched, fractal morphology to a more compact morphology (Mikhailov et al., 2006;Huang et al., 1994;Ramachandran and Reist, 1995;Weingartner et al., 1997;Martins et al., 1998). As the freshly emitted fractal

particles absorb liquid water due to the presence of hygroscopic species and become coated with organic material, capillary forces act to collapse the chain-like fractal particles. It is probable that most of the particles sampled during CARES are substantially aged and/or that the aging time (and subsequent collapse into compact shapes) is rapid. To quantify the extent



of particle compaction between T0 and T1, particle convexity ($Convexity = \frac{A}{A_{CvxHull}}$, where $A_{CvxHull}$ is the area of the convex hull around the particle area A) was calculated for the BC inclusions and is shown in **Figure 5**. Convexity distributions for inclusions from T0 and T1 are similar with the exception of a slightly higher number of inclusions with lower convexities at T0. Inclusions that are more branched have lower values of convexity (Coz and Leck, 2011), Hence, the

presence of more particles with lower convexities is consistent with the presence of fresh emissions at T0.

To compare the distribution of coating thicknesses at T0 and T1, the ratio of the BC core diameter ($D_{BC}$) to the total particle diameter ($D_{Total}$) was binned, scaled by the transmission efficiency and displayed in **Figure 6A**. Particles with thin coatings have $D_{BC}:D_{Total}$ approaching 1, whereas particles with thick coatings have $D_{BC}:D_{Total}$ approaching 0. The T0 site shows a slightly enhanced mode of thinly coated particles compared to T1, indicating the presence of fresh BC emissions at the T0

site. This mode of thinly coated particles follows the 1:1 line in the 2D histogram in **Figure 6C**. In a separate study, Bond et al. analyzed the various regions of the $D_{total}$ vs. $D_{BC}$ space (shown in **Figures 6B-C**), and found that particles that follow the 1:1 line are expected to have lower absorption amplifications compared to particles having thicker coatings (Bond et al., 2006). The majority of particles at both sites have thicker coatings; based on previous modeling studies these particles are expected to have larger absorption cross sections (Scarnato et al., 2013). As observed in other studies with stagnant regional

air masses (Moffet and Prather, 2009;Moffet et al., 2008), these results highlight the predominance of particles with thick coatings in source regions. For the CARES study, the source of these particles may be background transport from the neighboring industrial areas such as the San Francisco Bay Area.

**3.2 Location of BC Inclusions within Host Particle**

The location of BC inclusions within the particle affects the optical properties of the particle (Fuller et al., 1999). A previous
investigation associated with the CARES study found lower than expected absorption enhancements, possibly due to the location of the BC inclusions on the edge of the particles (Cappa et al., 2012). **Figure 7** shows the distribution of locations of the BC inclusions within their host particle for T0 and T1. To enable comparison of particles between all sizes, the distance ($R_{inc}$) of the BC inclusion center from the host particle center was normalized by the the longest distance ($R_{max}$) between the host particle center and the host particle edge (see graphic illustration in Fig. 7). In this case, a ratio of $R_{inc}/R_{max}=1$
corresponds to the longest distance of the BC inclusion from the particle center. Analysis of particles from two sampling sites showed no significant differences in the locations of the BC inclusions within host particles, suggesting that the distribution of BC inclusions does not vary substantially between the urban (T0) and rural (T1) sampling sites. **Figure 7** demonstrates that the majority of the BC inclusions were found in the center of their impacted host particle at both sites.

Of note, interpreting **Figure 7** may be biased because the distribution depends on the orientation of the BC inclusion shortly
before impaction. For example, a BC inclusion can be attached to the outside of the host particle at its lowest vertical coordinate (the bottom of the host). When impacted, the two dimensional image of the particle with the BC inclusion would appear to be at the center even though it was attached to the outside of the host particle. To explain the distributions in



**Figure 7** with respect to BC inclusion orientation, a model was constructed where the BC inclusion is positioned on or within spherical host (see supplementary section). From this model, calculated distributions of the BC inclusion location were derived assuming the particle was randomly distributed either on the surface or within the volume of the host. **Figure 7** demonstrates that the BC inclusions are preferentially located at the host center compared to the modeled distributions.

**Figure S3** demonstrates that if we change $R_{max}$ to be the radius of the largest circle inscribing the host particle, the BC inclusions measured near the center are still enhanced compared to the modeled distributions in **Figure 7**. The modeled distributions may under predict the number of BC inclusions in the center of the impacted host particle because 1) the BC inclusion is "pinned" to the surface while the host particle material spreads away from it, or 2) The BC inclusion is preferentially located at the host particle center due to the condensation of organic and inorganic material around the host particle in the atmosphere. To address point 1, a more detailed model of the particle impaction process is required. Specifically, the phase of the organic and inorganic material must be considered. If the inorganic/organic phase of the particles is liquid, it is possible that the impaction process can be modeled using computational fluid dynamics. However, there has recently been a body of research indicating that the organic material that frequently hosts BC inclusions may have a high viscosity, and thus resist spreading (Booth et al., 2014;O'Brien et al., 2014). Based on the analysis presented here, overall our data supports a coating model where the centrally located BC inclusion is covered by non-absorbing material. Such coated BC inclusions will have higher absorption than particles that are on the surface of the non-absorbing host.

## 4 Conclusions

The statistical analysis presented here represents the state of BC containing particles at source (T0) and receptor (T1) sites in the California central valley during a period of high photochemical activity and pollution buildup. During this period, the overall particle size at the receptor site was significantly larger due to the condensation of organic and inorganic species. The BC inclusion sizes between the T0 and T1 sites showed no detectable differences. The absorption efficiency of BC containing particles is strongly dependent on the size of the BC inclusions. Hence, measurements such as these in other geographical locations are important to understand the radiative impact of particles.

The extent of coating on individual particles was quantified by calculating the ratios of the BC inclusion area equivalent diameter to the host area equivalent diameter. The T0 site had a slightly enhanced population of thinly coated particles compared to the T1 site. This is consistent with the size distribution trends and the assumption that BC at the source site should have thinner coatings due to the presence of fresh BC emissions. The majority of the BC containing particles at both sites had thick coatings indicative of aged background particles. The high abundance of aged particles is consistent with the stagnant pollution plume present over the sampling site during this period. To model radiative transfer in aged urban pollution, particle-resolved measurements, such as those presented in this study, are valuable for characterization of the morphological properties and relative populations of aged vs. fresh emissions.



Previous particle resolved measurements of BC containing particles from the CARES campaign have highlighted the effects of BC inclusion/host geometry on absorption. Here, the location of the BC inclusions within the particles attached to the substrate was characterized. Using a model of particle impaction, it was shown that the BC inclusions were not all located at the center, nor were they all likely to be located on the surface of the particle prior to impaction. To improve our understanding of the location of the BC inclusion within the impacted particle, more accurate models of particle impaction and spreading are needed in future studies.

## 5 Acknowledgements

Funding for sample collection during CARES study was provided by the Atmospheric Radiation Measurement Program sponsored by the U.S. Department of Energy (DOE), Office of Science, Office of Biological and Environmental Research (OBER), Climate and Environmental Sciences Division (CESD). Funding for the data analysis was provided by the U.S. DOE's Atmospheric System Research Program, BER under grant DE-SC0008643. The STXM/NEXAFS particle analysis was performed at beamlines 11.0.2 and 5.3.2 at the Advanced Light Source (ALS) at Lawrence Berkeley National Laboratory. The work at the ALS was supported by the Director, Office of Science, Office of Basic Energy Sciences, of the U.S. DOE under contract DE-AC02-05CH11231. We thank A.L.D. Kilcoyne and T. Tyliszczak for their assistance with STXM experiments.



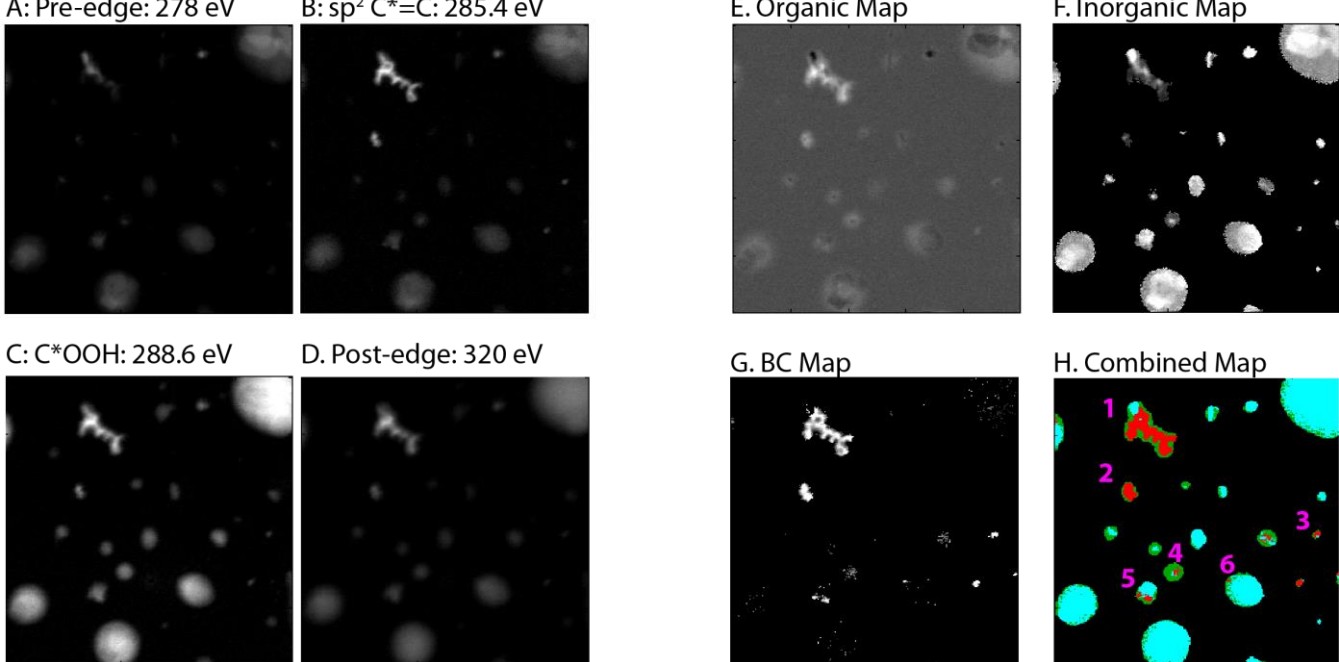

**Figure 1.** (A-C): Single energy images at 278, 285.4, 288.6, and 320 eV representing the pre-edge, C*=C $sp^2$ carbon, C*OOH, and total carbon. E) Organic map produced by subtracting the image at 278 eV from the image at 288.6 eV. F) Inorganic map produced by taking the ratio of the image at 278 eV to the image at 320 eV (Pre edge: total carbon) G) $sp^2$ carbon map derived by multiplying the ratio of 285.4 eV to 320 eV by a constant to give the percentage of $sp^2$ bonds as described in Hopkins et al. (Hopkins et al., 2007) H) Combined maps derived from thresholding of maps E-G; red areas contain %$sp^2$>35%, green areas are organic dominant and blue areas are non-carbonaceous inorganic dominant .





**Figure 2.** Cropped composition maps for all BC containing particles identified in CARES samples collected at T0 (top) and T1 (bottom). The submicron (left panels) particles are separated from the supermicron particles (right panels).





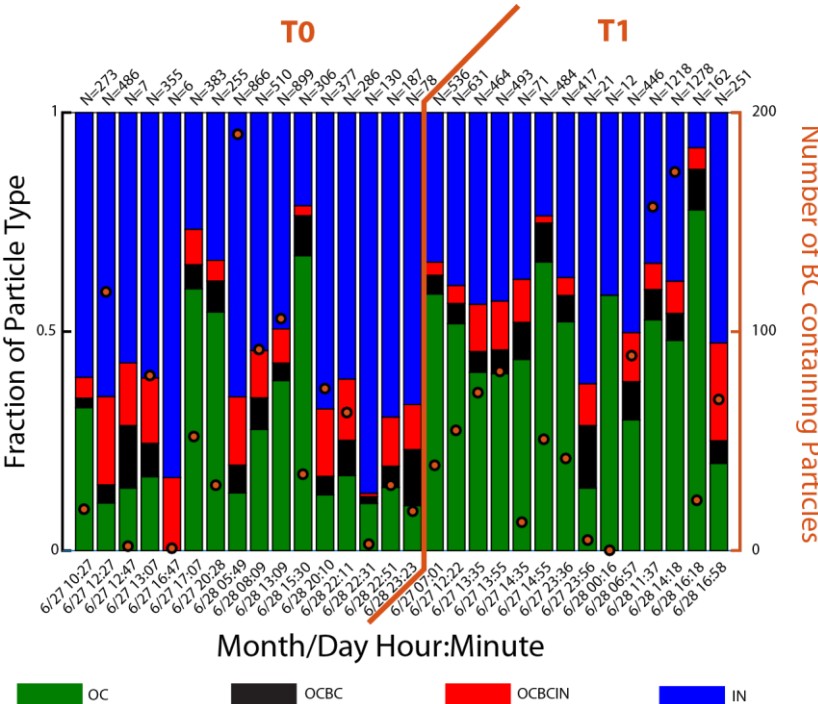

**Figure 3.** Summary of all particles analyzed from the CARES STXM data set. Green bars represent particles only containing the OC rich phase, black bars indicate particles containing BC and OC, red indicates the fraction of particles containing OC, BC, and IN and blue bars indicate the fraction of particles containing an inorganic dominant phase (IN). Orange dots indicate the total number of BC particles analyzed from a particular sample. The total number (N) of particles analyzed in each sample is indicated on top of the figure.





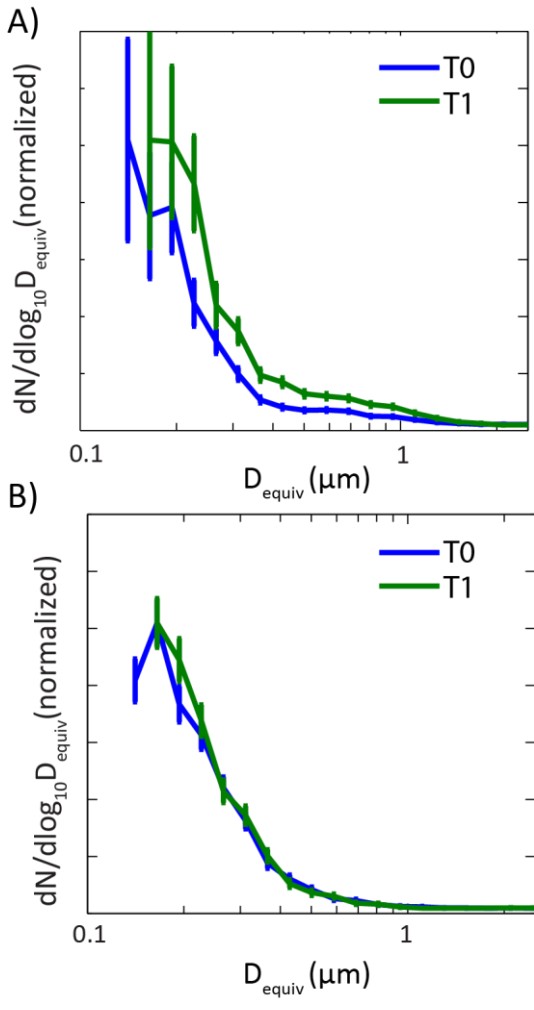

**Figure 4.** Panel A - Size distribution of BC containing particles at T0 (blue) and T1 (green). Panel B - Size distribution of BC inclusions at T0 (blue) and T1 (green).





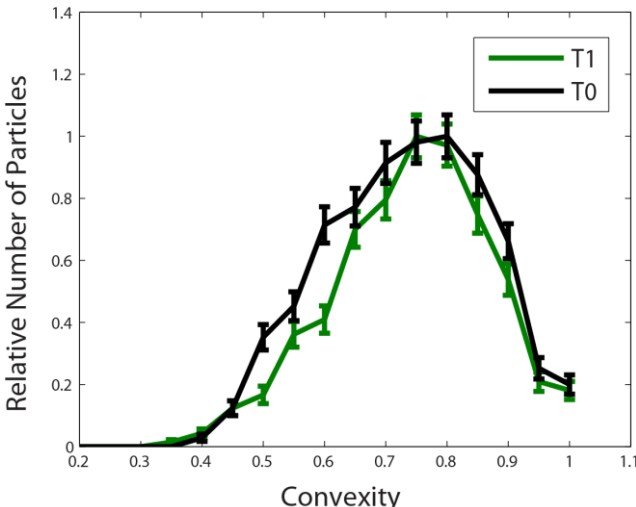

**Figure 5.** Histograms of BC inclusion convexity for T0 and T1 sites.

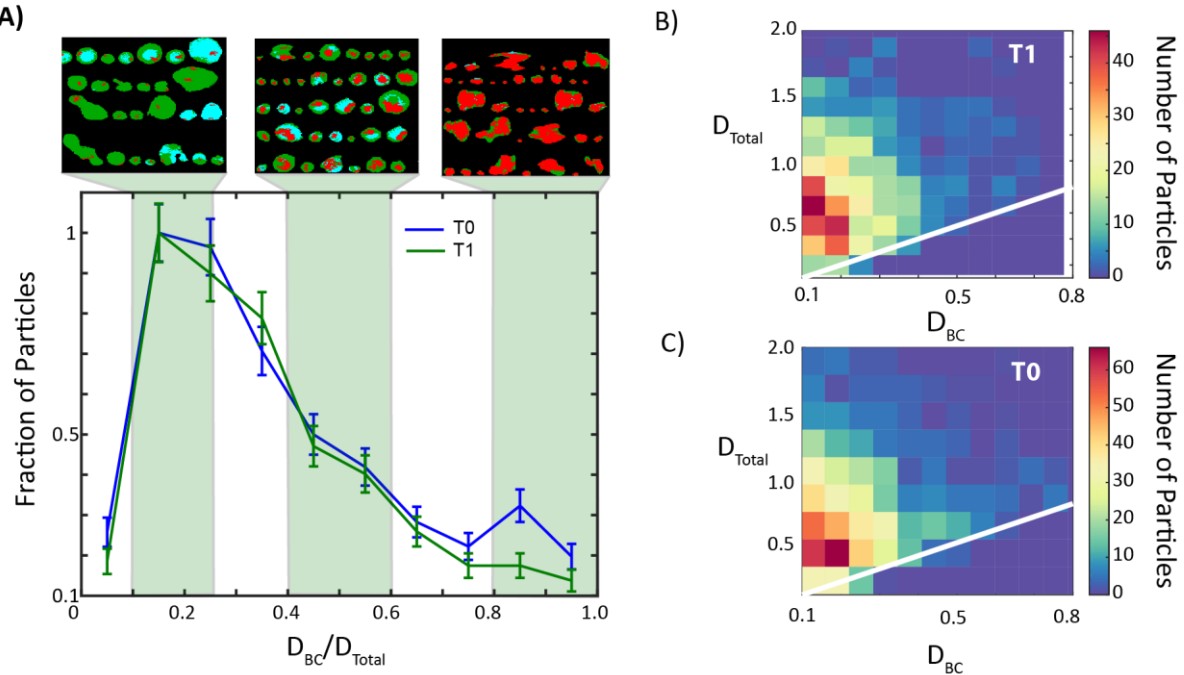

**Figure 6.** Panel A - Distributions of the ratio of the BC inclusion diameter to the total particle diameter ($D_{BC}:D_{total}$) after transmission efficiency correction. Images shown above the plot are examples of particles having (from left to right) $0.1 < D_{BC}:D_{total} < 0.25$, $0.4 < D_{BC}:D_{total} < 0.6$, $0.8 < D_{BC}:D_{total} < 1.0$. Panels B andC - Two dimensional histograms showing the number of particles having a total diameter $D_{total}$ and a BC core diameter $D_{BC}$ for T0 (panel C) and T1 (panel B). White lines indicate 1:1 ratios.





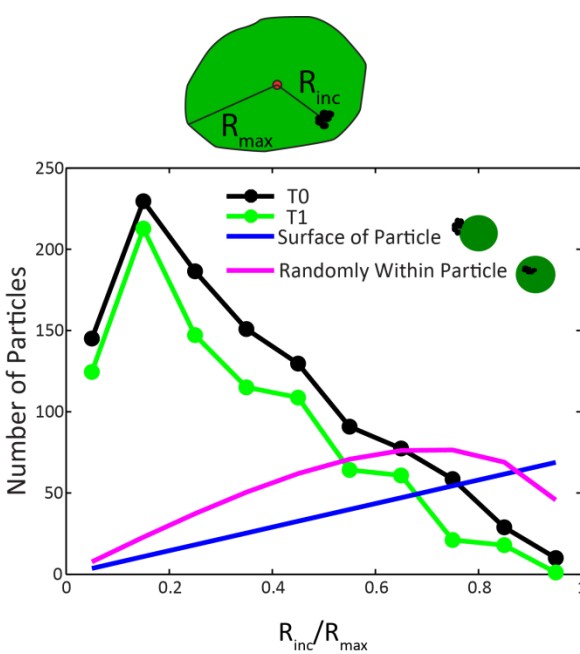

**Figure 7.** Distance of the BC inclusions from the center the host particle for the CARES field study at the T0 (black) or T1 (green) sites. The distance of the BC inclusion ($R_{inc}$) is normalized to the distance from the particle center of mass to the farthest edge ($R_{max}$) see illustration above plot). Modeled locations of BC inclusions are shown assuming the inclusion was on the surface (blue) or randomly within the host particle (magenta) before impaction.

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
