# Peer review of "Morphology and Mixing of Black Carbon Particles Collected in Central California During the CARES Field Study"

_Atmospheric Chemistry and Physics, 2016_

## Referee Comment (RC1) · Anonymous Referee #1 · 15 Aug 2016

This study shows mixing states of individual BC particles from relatively fresh and aged samples using STXM. The data is valuable for those who study aerosol mixing states and BC climate impacts. The technique used in this study is unique and includes important information of mixing states of BC, organic, and inorganic matters. The manuscript is well written, and I believe the manuscript contributes to the community.

P: Page L: Line

Specific comments 1: P1L23 "the contribution of fresh BC emissions at the urban site was relatively small". The sentence somehow contradicts to that in P6L17-19 "This result is expected considering that the T0 site is located in a source region for freshly emitted BC particles", P6L31-32 "It is probable that most of the particles sampled dur-

ing CARES are substantially aged and/or that the aging time (and subsequent collapse into compact shapes) is rapid" and P7L4-5 "Hence, the presence of more particles with lower convexities is consistent with the presence of fresh emissions at T0.". I suggest mentioning that "aging was rapid" or similar wordings.

2: P3L27: Please briefly describe how inorganic dominant region was obtained from the carbon edge regions in addition to the reference (Moffet et al., 2010). What kind of inorganic was detected here?

3: P3L28: Delete "to" from "to to".

4: P4L6: Please briefly describe how "35%" was used for the threshold. I believe the thresholds in STXM imaging are important to distinguish the materials.

5: P5L6: Please define (or refer the caption in Fig. 3) "OCBC, OCBCIN, OC, IN, INOC" in the main text in addition to the caption Fig. 3.

6: P6L26-27: "suggests that restructuring of the particles to more compact shapes upon transport is negligible" and Figure 5. I suggest discussing how useful to discuss the shape of BC using relatively low pixel size resolution (35 nm) comparing to BC monomer size (∼40 nm). Although the particle in Fig. 1 has fractal shape, most particles in Fig 2 do not show BC branches or BC monomers. I wonder if the BC images in Fig 2 is due to BC restructuring or a lack of image resolution.

7: P7L18: "3.2" should be "3.3"

8: P8L11: "Specifically, the phase of the organic and inorganic material must be considered" and Figures 2 and 7. This comment is a suggestion. In Fig. 2, it looks most BC locate inside of organic matter and outside of inorganic particles. It may be interesting to see if such difference is statistically true using the similar plot of Fig. 7.

9: P10 Figure 1. 9.1: Please add a scale bar. 9.2: In Figs.1C, E and H, the organic matter in E is larger than that in Fig.1C (COOH distribution), especially for particle 1. Is this artificial effect or real distribution? The BC Map (G) also looks similar enhancement

(e.g., images B vs G for particles 1 and 2). I think the choice of threshold relates to this issue. 9.3: Images E, F, and H: When focusing on relatively large inorganic dominant particles (e.g., right upper particle or left bottom particles), I see some inorganic rich inclusions coated by organic. However, in the combined map (H), I do not see such features but see organic only in the rim. Please explain what happen here.

10: P15L4: "see illustration above plot): "Take out ")" or add "(".
* * *

---

## Referee Comment (RC2) · Anonymous Referee #2 · 23 Aug 2016

The manuscript titled as "Morphology and Mixing of BC Particles Collected in Central California During the CARES Field Study" is about "characterization of the (BC) particle mixing state and morphology using Scanning Transmission X-ray Microscopy (STXM) at the carbon K-edge". In their work, characteristics of BC aerosol particles collected at two sampling sites were a main focus, which is worth to be published in ACP.

There are some specific comments which can help improve this manuscript.

(1) The description of sampling sites and conditions needs to be consistent and clearer in the manuscript. That is, in the experimental section, the two sampling sites were described as "the first site was in the Sacramento urban area (T0 site) expected to have enhanced fresh emissions (?), and the second site was located 40 km east of T0 in the Sierra Nevada foothills (T1 site) expected to have enhanced aged aerosol (?)." In the abstract and in the conclusion, it is said as "at both urban and rural sites" and "at source (T0) and receptor (T1) sites in the California central valley", respectively. Of course, these three different descriptions on the sampling sites may be related to each other, but without some concrete connections. It is better to make these descriptions consistent. In the experimental section, it is said that "Samples utilized here were collected over two days (June 27 and June 28) during a period of high temperatures and increased aerosol loadings over T0." And in the abstract and conclusion sections, it is said that "During a period of high photochemical activity and pollution buildup". I think these two description are not consistent. In addition, to relate the findings for BC particles collected at the two sampling sites, information on backward trajectories, sampling times and durations at T0 and T1 sites, and wind speed and direction needs to be given to better provide some clear idea about samples collected at T0 and T1 sites.

(2) In the abstract and Results section, thick "organic" coating is mentioned, and in the conclusion, it is said that "During this period, the overall particle size at the receptor site was significantly larger due to the condensation of organic and inorganic species". Indeed, I am curious about the modification of "inorganic" species during the possible aging process. If some argument about inorganic species aging is given in the Results section, it will be interesting.

(3) The abstract and conclusion parts need to be rewritten to convey the findings and meaning of this work more consistently and clearly.

(4) List of Awkward and/or ambiguous sentences and/or sections

- p. 3, lines 22-23

- p. 4, lines 4-6 (and needs to say why)

- p.4, lines 9-21: only the expert of this technique could understand this part.

- p.6, lines 3-6: Difficult to understand.

- p.6, line 20: droplet mode → ???

- p.6, lines 31-31 and p.7, lines 4-5: This description is mostly inconsistent with some descriptions given elsewhere in the manuscript.

- p.7, line 16: "in source regions" and "the source of these particles" → ???

- p.8, lines 25-29: Hard to understand.

(5) Typos:

- P.3, line 8 : relative humidity → moisture

- P.3, line 13 : definition of rho is missing.

- P. 4, line 24 : molecular markers → chemical (or functional group) markers

- P.4, line 32

- P. 5, line 29: maps OCBC and OCBCIN, particles → maps, OCBC and OCBCIN particles

- P.5, line 30: INOC →IN

- Many places in the manuscript: $D_{BC}:D_{Total}$ → $D_{BC}/D_{Total}$

- P.10, Figure 1 caption : A-C → A-D

- P.11, Figure legend: Soot inclusion → BC inclusion

- P.14, Figure caption : $D_{BC}:D_{Total}$ → $D_{BC}/D_{Total}$

---

## Author Comment (AC1) · 27 Oct 2016

**RED=REVIEWER COMMENT**
**GREEN=AUTHOR RESPONSE**

This study shows mixing states of individual BC particles from relatively fresh and aged samples using STXM. The data is valuable for those who study aerosol mixing states and BC climate impacts. The technique used in this study is unique and includes important information of mixing states of BC, organic, and inorganic matters. The manuscript is well written, and I believe the manuscript contributes to the community.

We thank the reviewer for taking the time to comment on our manuscript. We have taken the reviewers comments into account in a revised version of our manuscript. Below we detail how each comment was addressed in the revised manuscript.

**P: Page L: Line**
Specific comments 1: P1L23 "the contribution of fresh BC emissions at the urban site was relatively small". The sentence somehow contradicts to that in P6L17-19 "This result is expected considering that the T0 site is located in a source region for freshly emitted BC particles", P6L31-32 "It is probable that most of the particles sampled during CARES are substantially aged and/or that the aging time (and subsequent collapse into compact shapes) is rapid" and P7L4-5 "Hence, the presence of more particles with lower convexities is consistent with the presence of fresh emissions at T0.". I suggest mentioning that "aging was rapid" or similar wordings.

We have added the possibility of rapid aging to the abstract;
Page 1, Line 23: "Based on the observation of thick coatings and more convex BC inclusion morphology, either the aging was rapid or the contribution of fresh BC emissions at the urban site was relatively small."

2: P3L27: Please briefly describe how inorganic dominant region was obtained from the carbon edge regions in addition to the reference (Moffet et al., 2010). What kind of inorganic was detected here?

To address this comment, we have added some detail to the revised manuscript:

Page 3-4 Lines 33-2 The inorganic map, representing non-carbonaceous inorganic dominant regions, is derived from the ratio of the pre edge (278 eV) to the post-edge (320 eV) ratio (ODpre/ODpost) and is shown in **Figure 1F**. Non-carbonaceous, inorganic inclusions of $(NH_4)_2SO_4$ and NaCl were confirmed using energy dispersive X-ray spectroscopy for these CARES samples (Moffet et al., 2013).

We also refer the reviewer to our response to comment #4 below in which we include more details about thresholding. Page 4 Lines 15-18

3: P3L28: Delete "to" from "to to".

Corrected.

4: P4L6: Please briefly describe how "35%" was used for the threshold. I believe the thresholds in STXM imaging are important to distinguish the materials.

We have addressed this request by adding more specific detail about how the thresholding was carried out. These changes also address reviewer concern #2. The revised wording is as follows:

Page 4 Lines 11-18 "Thresholds for each of the three maps in Figure 1E-G were set using the following criteria: 1) pixels at 288.6 eV with intensities three times below signal to noise ratio were set to zero 2) pixels having ODpre/ODpost<0.5 were set to zero as discussed in Moffet et al. 2010 and 3) the %sp2 was set to zero below a value of 35%. The empirical determination of the threshold value of 35% is discussed elsewhere (Moffet et al., 2011). Areas of each of the maps with fewer than 7 conjoined pixels were excluded. Thresholds for maps in Figure 1E-1G were applied to produce binary images. In Figure 1H, to highlight the BC mixing state these three binary images are overlaid in the following order: organic, inorganic, sp$^2$."

5: P5L6: Please define (or refer the caption in Fig. 3) "OCBC, OCBCIN, OC, IN, INOC" in the main text in addition to the caption Fig. 3.

We have added these definitions to Page 5 Line 8 of the revised manuscript:

Page 5 Line 8 "Based on these maps, the mixing state of the particles was stored in a database for the subsequent analysis of BC morphology. The label-based mixing state is defined from particles having organic carbon (OC), black carbon (BC), or inorganic (IN) regions defined by the binary maps shown in **Figure 1H**. For example, if a particle contains both organic and black carbon regions it is labeled OCBC and so on."

6: P6L26-27: "suggests that restructuring of the particles to more compact shapes upon transport is negligible" and Figure 5. I suggest discussing how useful to discuss the shape of BC using relatively low pixel size resolution (35 nm) comparing to BC monomer size (_40 nm). Although the particle in Fig. 1 has fractal shape, most particles in Fig 2 do not show BC branches or BC monomers. I wonder if the BC images in Fig 2 is due to BC restructuring or a lack of image resolution.

In order to address this comment, as well as to add the possibility that resolution affects our ability to detect restructuring, we have added the following text:

Page 7 Line 20 The resolution of the STXM instrument limits the ability to identify small (<100 nm) branched/fractal inclusions. However, even freshly emitted soot particles become more compact at smaller sizes (Park et al., 2004). Moreover, monomer size tends to be around 40 nm, which should be detectable by the STXM instrument, though operating at the very limit. The monomers of BC aggregates are typically connected in order to form a branched, fractal particle, resulting in the observation that only larger particles have high fractal dimension.

We have added the following reference to support this added discussion:

Park, K., Kittelson, D. B., and McMurry, P. H.: Structural properties of diesel exhaust particles measured by transmission electron microscopy (TEM): Relationships to particle mass and mobility, Aerosol Sci Tech, 38, 881-889, 10.1080/027868290505189, 2004.

7: P7L18: "3.2" should be "3.3"

Corrected

8: P8L11: "Specifically, the phase of the organic and inorganic material must be considered" and Figures 2 and 7. This comment is a suggestion. In Fig. 2, it looks most BC locate inside of organic matter and outside of inorganic particles. It may be interesting to see if such difference is statistically true using the similar plot of Fig. 7.

We felt that this was an important point that is related to the observation made on P5L14-15 of the original manuscript: "Frequently, BC inclusions were seen on the outside edge of an inorganic dominant region; this arrangement may have occurred upon efflorescence, when a salt excludes the aqueous phase (Liu et al., 2008)."

In order to address this hypothesis in a more quantitative fashion, we observed that the radial distribution of BC inclusions of OCBCIN particles has a significantly higher fraction of BC inclusions closer to the edge of the particle. Particles that contained large IN (inorganic) inclusions had an even higher proportion of BC inclusions closer to the edge of the host particle. We have added these distributions to Figure 7 along with the following discussion in the results section:

Page 8 Line 16 of the revised manuscript:

 "Analysis of particles from two sampling sites showed minor differences in the locations of the BC inclusions within host particles, suggesting that the distribution of BC inclusions does not vary substantially between the urban (T0) and rural (T1) sampling sites. Slightly more BC inclusions were found closer to the edge of the host particle at T0. This is likely due to the higher frequency of inorganic inclusions at T0; the crystalline inorganic phase may push the BC inclusions away from the center upon efflorescence. Moreover, BC particles may more easily mix with the OC phase when the particle is in the dry state. The bottom panel of **Figure 7** demonstrates that particles containing inorganic rich phases (OCBCIN particles) have an enhanced number of particles with the BC inclusion near the edge of the host; this trend is enhanced when particles with large (500 nm) inorganic inclusions are considered. These results demonstrate that the majority of the BC inclusions were found in the center of their impacted host particle at both sites and that the presence of inorganic dominant inclusions acts to push the BC inclusion farther from the center of the host."

A sentence was added to the conclusions of the revised manuscript (Page 9-10:Lines 32-1):

"Particles containing inorganic rich inclusions were more likely to have the BC inclusion pushed towards the edge of the host."

The following was added to the abstract (Page 1,Line 27):

Most particles were observed to have the BC inclusion close to the center of the host particle. However, hosts containing inorganic rich inclusions had the BC inclusion located closer to the edge of the particle.

9: P10 Figure 1. 9.1: Please add a scale bar.

We have added a scale bar.

9.2: In Figs.1C, E and H, the organic matter in E is larger than that in Fig.1C (COOH distribution), especially for particle 1. Is this artificial effect or real distribution? The BC Map (G)

also looks similar enhancement C2(e.g., images B vs G for particles 1 and 2). I think the choice of threshold relates to this issue.

For particle 1, it is likely that the organic matter appears to be larger in E due to the fact that the map is derived from more than one image and the contrast is different. Images A-D are single energy images with large differences in absorbance values due to differences in thickness. For example, comparing B vs. G is difficult/non-trivial because B is a single energy image whereas the "BC Map" (G) is derived by the ratio of image B divided by the total carbon (Image D – Image A). Thus, Image B is dependent on thickness, whereas map G is independent of thickness. There is reason to believe that they are different because they are representing two physically different measurements.

9.3: Images E, F, and H: When focusing on relatively large inorganic dominant particles (e.g., right upper particle or left bottom particles), I see some inorganic rich inclusions coated by organic. However, in the combined map (H), I do not see such features but see organic only in the rim. Please explain what happen here.

This is a threshold issue. Even though the rims are less enriched in inorganics, the $OD_{pre}/OD_{post}$ is still greater than 0.5, indicating inorganic rich coatings.

10: P15L4: "see illustration above plot): "Take out ")" or add "(".

We have made the correction.

---

## Author Comment (AC2) · 27 Oct 2016

RED=REVIEWER COMMENT
GREEN=AUTHOR RESPONSE

The manuscript titled as "Morphology and Mixing of BC Particles Collected in Central California During the CARES Field Study" is about "characterization of the (BC) particle mixing state and morphology using Scanning Transmission X-ray Microscopy (STXM) at the carbon K-edge". In their work, characteristics of BC aerosol particles collected at two sampling sites were a main focus, which is worth to be published in ACP.

We thank the reviewer for their comments. Below we provide our detailed responses.

There are some specific comments which can help improve this manuscript.

1. The description of sampling sites and conditions needs to be consistent and clearer in the manuscript. That is, in the experimental section, the two sampling sites were described as "the first site was in the Sacramento urban area (T0 site) expected to have enhanced fresh emissions(?), and the second site was located 40 km east of T0 in the Sierra Nevada foothills (T1 site) expected to have enhanced aged aerosol(?)."In the abstract and in the conclusion, it is said as "at both urban and rural sites" and "at source (T0) and receptor (T1) sites in the California central valley", respectively. Of course, these three different descriptions on the sampling sites may be related to each other, but without some concrete connections. It is better to make these descriptions consistent.

We believe that the prevalence of aged aerosol at the urban site is perhaps contrary to expectations, so we have highlighted this by changing the sentence in the abstract (Page: 1 Line: 21-23 of the revised manuscript) to read:

"Observations of compacted BC core morphologies and thick organic coatings at both urban and rural sites provide evidence of the aged nature of particles, highlighting the importance of highly aged particles at urban sites during periods of high photochemical activity."

In the experimental section, it is said that "Samples utilized here were collected over two days (June 27 and June 28) during a period of high temperatures and increased aerosol loadings over T0."And in the abstract and conclusion sections, it is said that "During a period of high photochemical activity and pollution buildup". I think these two description are not consistent.

As the reviewer has suggested, we have made the connection between "pollution buildup" and "photochemical activity" more explicit. Specifically:

The sentence on Page: 3 Line: 6 of the revised manuscript was changed to read:

"Samples utilized here were collected over two days (June 27 and June 28) during a period of high temperatures and increased aerosol loadings over T0 due to high photochemical activity."

In addition, to relate the findings for BC particles collected at the two sampling sites, information on backward trajectories, sampling times and durations at T0 and T1 sites, and wind speed and direction needs to be given to better provide some clear idea about samples collected at T0 and T1 sites.

As the referees suggested, we have added a more detailed discussion of meteorological conditions during this episode during the CARES field study. Meteorology was covered extensively in the Fast et al. reference and details of CO modeling were further presented in the Moffet 2013 reference. We have elaborated on these points in the experimental section (Page: 3 Line: 8):

"CO tracer modeling indicated that significant transport from the San Francisco Bay Area affected the Sacramento site in the morning while the boundary layer was low. Later on in the day a larger fraction of emission at the T0 site were from the Sacramento metropolitan area. Similar contributions from the San Francisco Bay Area were modeled at T1, however, emissions from Sacramento constituted the largest source of emissions during this time period (Moffet et al., 2013; Fast et al., 2014)."

2. In the abstract and Results section, thick "organic" coating is mentioned, and in the conclusion, it is said that "During this period, the overall particle size at the receptor site was significantly larger due to the condensation of organic and inorganic species". Indeed, I am curious about the modification of "inorganic" species during the possible aging process. If some argument about inorganic species aging is given in the Results section, it will be interesting.

Although aging of inorganic species (NaCl reacting with acids, oxidation $SO_2$ to $SO_4$ etc.) was not the focus of this manuscript, discussion of the aging of inorganic rich particles through the process of coagulation with BC can be found here:

Revised manuscript Page: 5 Line: 21: The most striking difference between BC particles from T0 and T1 is the high amount of inorganic dominant regions for T0 particles. T0 was impacted by sea spray and sulfates from petroleum refineries located in the San Francisco Bay Area (Laskin et al, 2012). The large inorganic dominant particles can be attributed to sea spray particles that have coagulated with BC emissions from the Bay Area.

3. The abstract and conclusion parts need to be rewritten to convey the findings and meaning of this work more consistently and clearly.

This relates to point #1, and we have modified the abstract and results to make the connection between pollution buildup and photochemical activity more clearly and consistently.

**(4) List of Awkward and/or ambiguous sentences and/or sections**

-p. 3, lines 22-23

We understand that the reviewer may have found this awkward, so we have reworded the sentence to read:

Page 3, Line 25: "To generate a carbon-based map, aerosol particles were imaged at 278 eV (the carbon pre-edge), 285.4eV (sp$^2$ C*=C), 288.6eV (C*OOH), and 320 eV (the carbon post-edge). Characteristic single energy images at these energies are shown in **Figure 1A-D**. Typically, at each energy, a 15 x15 µm$^2$ image was acquired with 0.035 µm pixel size and 1 ms dwell time. Occasionally, ~120 different constant energy images were utilized in this study to obtain a high resolution carbon spectrum. For consistency, the same set of four constant-energy images (278, 285.4, 288.6, and 320 eV) were used to characterize particles for this analysis."

-p. 4, lines 4-6 (and needs to say why)

We have stated the purpose in the beginning of the sentence:

"To define organic carbon rich (OC), inorganic non-carbonaceous rich (IN), and black carbon (BC) regions, thresholds for each of the three maps in Figure 1E-G were set using the following criteria:"

-p.4, lines 9-21: only the expert of this technique could understand this part.

We have attempted to make this section more accessible to the layperson – however, we feel that while only an expert can understand parts of this section, the information is necessary to provide a more complete documentation of the analysis. Much of this paragraph outlines quality control measures. To make this clear, we have added an introductory sentence:

Page 4 Line 21 of revised manuscript: "A brief description of quality control measures is summarized here."

-p.6, lines 3-6: Difficult to understand.

We have added an introductory explanation of the concept of diversity – which we suspect is what the reviewer found difficult to understand:

Page 6 Line 15 of revised manuscript: "In O'Brien et al. (2015), entropy metrics (Riemer and West, 2013) were used to calculate a diversity that represents the effective number of species per particle or in the bulk population. In this case diversities were specified using the OC, BC, and IN components such that a particle or population can have a maximum diversity of 3."

-p.6, line 20: droplet mode ☐???

We have added the approximate range of the droplet mode from 200 – 1000 nm.

-p.6, lines 31-31and p.7, lines 4-5: This description is mostly inconsistent with some descriptions given elsewhere in the manuscript.

We understand the reviewer believes that the presence of fresh emissions is contradictory to the statements in the abstract and the results and discussion. We have carefully worded the results in the abstract to state that there was a small population of fresh emission detected at T0. To emphasize the relative contributions of fresh and aged particles we have added the following discussion to Page 7 Line 4 of the revised manuscript:

"…comparison of the convexity distributions between T0 and T1 indicates a small statistically significant population of less compact particles at T0 consistent with fresh emissions. However, it should be emphasized that the majority of particles at both sites have compact shapes and are likely due to the prevalence of aged BC containing particles."

-p.7, line 16: "in source regions"and "the source of these particles"☐???

We changed "source regions" to "urban areas with similar meteorological conditions" as that more accurately conveys the main idea of the sentence.

-p.8, lines 25-29: Hard to understand.

Again, this relates to the seemingly contradictory concept that we observe fresh particles at T0 while it is emphasized elsewhere that most of the BC particles are highly aged. To address this we have emphasized that the population of thinly coated particles is small as well as other minor clarifying edits:

Page 9, Line 22 of the revised manuscript: "The T0 site had a small population of thinly coated particles compared to the T1 site. This is consistent with slightly smaller overall particle sizes at T0 and the assumption that BC at the source site (T0) should have thinner coatings due to the presence of fresh BC emissions."

(5) Typos:

- P.3, line 8 : relative humidity ->moisture

corrected

- P.3, line 13 : definition of rho is missing.

corrected

- P.4, line 24 : molecular markers ->chemical (or functional group)markers

corrected

- P.4, line 32

Did not find a typo

- P.5, line 29: maps OCBC and OCBCIN, particles ->maps, OCBC and OCBCIN particles

corrected

- P.5, line 30: INOC->IN

Added "particle types" to the end of the sentence to convey our original meaning.

- Many places in the manuscript: DBC:DTotal->DBC/DTotal

corrected

- P.10, Figure 1 caption : A-C ->A-D

corrected

- P.11, Figure legend: Soot inclusion ->BC inclusion

corrected

- P.14, Figure caption : DBC:DTotal->DBC/DTotal

corrrected